# Effect of Intravenous Iron Administration on Bone Mineral and Iron Homeostasis in Patients with Inflammatory Bowel Disease—Results of a Prospective Single-Centre Study

**DOI:** 10.3390/jpm13030458

**Published:** 2023-02-28

**Authors:** Edyta Tulewicz-Marti, Paulina Szwarc, Martyna Więcek, Konrad Lewandowski, Tomasz Korcz, Malgorzata Cicha, G. Rydzewska

**Affiliations:** 1Clinical Department of Internal Medicine and Gastroenterology with Inflammatory Bowel Disease Subunit, National Medical Institute of the Ministry of the Interior and Administration, 02-507 Warsaw, Poland; 2Laboratory Diagnostics Department, National Medical Institute of the Ministry of the Interior and Administration, 02-507 Warsaw, Poland; 3Collegium Medicum, Jan Kochanowski University, 25-317 Kielce, Poland

**Keywords:** ironformula, IBD, inflammatory bowel disease, hipophosphatemia, vitamin D

## Abstract

Introduction: Anaemia and bone metabolism alterations are common in inflammatory bowel disease (IBD), which is a heterogeneous group of diseases that include Crohn’s disease (CD) and ulcerative colitis (UC) with a rich intestinal and extraintestinal symptomatology. All these make the diagnostic procedures complicated and difficult. Purpose and scope: The aim of this study was to assess the effect of parenteral iron administration on biomarkers of mineral and bone homeostasis over time. Materials and methods: The study was a single-centre non-randomised prospective study. It was carried out between 2016 and 2020 in a group of patients in the Department of Internal Medicine and Gastroenterology Subunit of Inflammatory Bowel Diseases at the National Institute of Medicine of the Ministry of the Interior and Administration in Warsaw. At the first examination, the baseline disease severity, initial evaluation of anaemia (morphology, iron (Fe), total iron binding capacity (TIBC), ferritin, vitamin B_12_, folic acid) and bone mineral metabolism including C-reactive protein (CRP), albumins, alkaline phosphatase (ALP), Calcium, osteocalcin, phosphate in serum and in urine, parathyroid hormone (PTH), vitamin D3, fibroblast growth factor (iFGF23) and procollagen type 1N propeptide (P1NP) C-terminal telopeptide (CTX), was initially assessed. On the basis of peripheral blood counts, an appropriate dose of iron (iron derisomaltose or caboxymaltose) was administered. During the subsequent appointments on week 1, 4, and 12 morphology, iron (Fe), total iron binding capacity (TIBC), ferritin, vitamin B12, folic acid, C-reactive protein (CRP), albumins, alkaline phosphatase (ALP), Calcium, osteocalcin, phosphate in serum and in urine, parathyroid hormone (PTH), vitamin D3, fibroblast growth factor (iFGF23) and procollagen type 1N propeptide (P1NP) C-terminal telopeptide (CTX), were evaluated. Results: A total of 56 patients were enrolled into the study: 24 women and 32 men. In the group, 32 patients had Crohn’s disease (CD) and 24 had ulcerative colitis (UC). We found a statistically significant increase in the concentration of albumin (*p* = 0.031), haemoglobin (*p* < 0.001), haematocrit (*p* < 0.001), MCV (*p* < 0.001), MCHC (*p* = 0.001), iron (*p* < 0.001) and ferritin (*p* < 0.001) after the administration of parenteral iron. The influence of individual iron formulations on the analysed parameters (phosphate concentration in serum and in the urine, iFGF23, P1NP, PTH, vitamin D, haemoglobin and ferritin) was similar. Interestingly, an inverse correlation was found between the concentration of phosphorus in the blood and iFGF23 at certain time-points; however, in the study group they did not significantly affect the disturbances of calcium and phosphate metabolism. Conclusions: In the study group, transient and non-significant disorders of phosphate metabolism were found, which does not constitute a contraindication to treatment with parenteral iron in inflammatory bowel disease patients, which was safe and efficient.

## 1. Introduction

Inflammatory bowel disease (IBD) is a heterogeneous group of diseases that includes Crohn’s disease (CD) and ulcerative colitis (UC) and it is characterized by a rich intestinal and extraintestinal symptomatology, including iron deficiency anaemia and bone metabolic disease [1,2,3,4]. Anaemia is a common complication of IBD; in adults with IBD, its prevalence may affect up to almost half of the IBD population [5] and as much as three fourths of those who are hospitalised [6,7,8]. It is considered an inevitable complication of the disease and iron deficiency anaemia (IDA) was identified as the most common type but still it is an underestimated problem [9]. IDA has a significant negative impact on patients’ quality of life and it was observed that iron repletion may improve it [6,7]. Typical symptoms of IDA are fatigue, headache, vertigo, or tachycardia and they may not be easily distinguished from symptoms of IBD. Less common symptoms include restless leg syndrome or reduced cognitive and physical performance. 

Given the fact that oral iron formulations may be badly tolerated by IBD patients, the European Crohn’s and Colitis Organisation (ECCO) guidelines state that iron deficiency and IDA should be treated with high-dose IV iron [8]. Even though there is some data showing that in mild or moderate anaemia in IBD oral formulations may be useful, in general administered intravenously iron is more effective, delivers a faster response and is better tolerated than oral iron [10,11]. Despite its effectiveness, the anaemia may recur in IBD patients (median: 19 months for iron deficiency and 10 months for IDA), which makes repeated infusions necessary. The most widely used IV iron formulations in Europe are ferric carboxymaltose (Ferinject®) and iron derisomaltoside (Monover^®^). These medications are safe, though hypophosphatemia is a well-known side effect of both intravenous preparations [12,13]. Data from clinical trials and from literature indicate that some of these medications may be more likely to cause hypophosphatemia than others and may be related with elevation of intact fibroblast growth factor -23 (iFGF-23). 

Phosphate is essential in human physiology [14,15] and a deficiency (0.65 mmol/L) may cause symptoms like fatigue, proximal muscle weakness and bone pain [11], being difficult to distinguish from IBD symptoms. Furthermore, prolonged hypophosphatemia can result in osteomalacia. There is little data on the clinical manifestations of IV iron in IBD patients and its influence on bone mineral homeostasis. 

The aim of this study was to investigate changes in calcium and phosphate levels in blood and urine, iFGF 23 after iron infusions of carboxymaltose and derisomaltose in adults with IBD. 

## 2. Materials and Methods

### 2.1. Study Design and Patient Population 

#### Study Population

We conducted a non-randomised single-centre prospective study. Patients with IBD were prospectively recruited in the Department of Internal Medicine and Gastroenterology’s IBD Subunit at the Central Clinical Hospital in Warsaw, Poland. The inclusion criteria were an age of 18 years or more, a verified diagnosis of IBD based on clinical, endoscopic, biochemical and histological findings, a minimum of 6 months from diagnosis and patients who needed iron supplementation according to ECCO guidelines and were able to read and understand Polish and give written consent. Disease activity was evaluated by the Crohn’s Disease Activity Index (CDAI) in patients with Crohn’s disease and the Truelove Score in those with UC. A CDAI score of less than 150 points was defined as remission; between 150 and 220 was low disease activity; between 220 and 450 points moderate disease activity; and more than 450 high. We excluded from the study patients who had received a blood transfusion in the last 2 months, those who had received oral supplementation in the last 3 months, those with a severe comorbidity such as chronic kidney disease, thyroid, or parathyroid gland disease, those with osteoporosis or new calcium or vitamin supplementation, those with a severe infection such as *Clostridioides difficile* and those who did not cooperate with the clinician. The inclusion period lasted from 2016 until 2020. On the basis of peripheral blood counts, an appropriate dose of iron (ferric derisomaltoside- Monover, FDI) or in case of intolerance or unresponsive to FDI, iron carboxymaltose (Ferinject, FCM) was administered. Patients who during the iron infusion presented serious adverse events were excluded from the study. During the subsequent appointments on day 7, in week 4, and week 12, the following parameters were measured in the patients: in the peripheral blood serum – blood count, phosphate in serum, calcium, ALP, iFGF23, vitamin D_3_, PTH, CRP, Cr, albumin, iron, TIBC, ferritin, osteocalcin, P1NP and C-telopeptide; in the urine – phosphate; and in the stool – calprotectin. Disease activity was evaluated by the disease activity form. 

### 2.2. Clinical, Sociodemographic and Laboratory Variables

Data such as gender, type of disease, previous operations and current medications were collected by interviews and from medical records at enrolment. The demographic and clinical data are shown in Table 1. 

All laboratory data analysis was performed at the local laboratory. A C-protein (CRP) level of 5 or higher was chosen to indicate active inflammation. Samples were collected in the morning between 8.00 and 11.00 am after an overnight fast. A urine sample was obtained in the same day. The blood was allowed to clot, then separated and centrifuged at 4 °C; it was finally frozen at −40 °C before processing. Laboratory investigations were measured by routine hospital laboratory methods and included serum albumin, alkaline phosphatase, calcium, phosphate, CRP, serum 25-(OH)-D and bone markers and phosphate in urine. 

### 2.3. Statistical Analysis

BM SPSS Statistics 25 was used to answer the research question. Descriptive statistics, together with Kolmogorov-Smirnov tests, were calculated. The U Mann–Whitney test, Friedman test, chi-square test of independence and Pearson’r correlation was performed. *p* < 0.05 was considered statistically significant.

### 2.4. Ethical Considerations

The Ethical Committee of the National Institute of Medicine (previously Central Clinical Hospital) of the Ministry of the Interior and Administration approved this study (28/2016).

## 3. Results

Of the 56 patients, 44 received ferric derisomaltoside and 12 received ferric carboxymaltose. The changes in biological indicators were analysed between the 3 measurements: before drug administration and 1 week, 4 weeks and 12 weeks after administration. There was a statistically significant increase in the concentration of haemoglobin (*p* < 0.001), iron (*p* < 0.001) and ferritin (*p* < 0.001) after the administration of parenteral iron (Figure 1).

There was a trend towards a decreasing phosphate concentration at week 4, followed by an increase at week 12, but this was not statistically significant (*p* = 0.054). There were no statistically significant differences in the remaining parameters tested, such as phosphorus in urine and iFGF23, at the selected time-points There were no statistically significant differences in the examined disorders of calcium and phosphate metabolism and bone turnover markers (Figure 2 and Figure 3).

The change in biological markers between the 1st measurement and the measurement on day 7 after drug administration was assessed, and also the differences in the biological markers 7 days after drug administration. Detailed data are shown in Table 2.

There were no significant differences in measurements in patients receiving iron formulations (Table 3). 

The analysis did not reveal any significant relationships between changes in biological indicators and the type of drug taken. The changes in individual biological indices were similar in subjects taking derisomaltose or ferric carboxymaltose (Table 3).

### Relationship of Phosphate and iFGF23 Levels in Individual Measurements

The relationship of phosphate level and iFGF23 in each of the 3 measurements was analysed. Detailed data are presented in Table 4.

A statistically significant correlation was demonstrated in weeks 4 and 12 between the decrease in phosphate concentration and the increase in iFGF23. The correlation was clearly visible in the ferric carboxymaltose group (Table 5). Although there was a relationship between these parameters, in the study group they did not significantly affect the calcium and phosphate metabolism disturbances (Table 6).

## 4. Discussion

In this non-randomised single-centre prospective study on anaemic patients with IBD, we found that both iron formulations were safe and effective in treating anaemia in IBD patients. 

Our study has many strengths. Firstly, it evaluated the efficacy and security of both intravenous iron formulas in IBD patients, which is an important issue given the fact that up to 95% of patients may be at risk of bone mineral alterations [16,17,18,19,20,21]. Jahnsen et al. have shown that hypovitaminosis in IBD patients is common and that patients with CD and after small-bowel resections are especially at risk of developing secondary hyperparathyroidism and low BMD [20]. Previous clinical trials’ data suggest higher risk for the development of hypophosphatemia associated with FCM [13], however higher risk of mild hypersensitivity reactions were found in FDI group [20]. Moreover, as reported in recent review and metanalysis hypophosphatemia may persist at the end of the study periods (maximum 3 months) in up to 45% of patients treated with FCM [22]. According to our study, iron formulas were safe. Even though after iron repletion in the 4th week trend towards a decrease of phosphate was observed in our patients it was transient and not statistically significant. Looking more closely, increase and normalization of phosphate levels was observed in the 12th week of observation. As it was previously shown in population of patients with IBD, both formulations are safe and effective in this group of patients [23,24,25]. 

Secondly, in order to check the possible mechanism of hypophosphatemia and its relationship with particular iron formulations, we analysed iFGF23, which is a hormone produced by the osteocytes which increases the rate of urinary excretion of phosphate and inhibits 1,25-dihydroxyvitamin D production [26]. We found a statistically significant correlation in weeks 4 and 12 between the decrease in phosphate concentration and the increase in iFGF23, especially in the FCM group. Although there was a relationship between these parameters, they did not significantly affect the calcium and phosphate metabolism disturbances. This is consistent with the data from Dahlerup et al., who also did not find any severe disturbances regarding iFGF2 in patients who received iron derisomaltose [25]. In the clinical trial performed by Wolf et al. comparing influence of intravenous ferric derisomaltose vs ferric carboxymaltose on hypophosphatemia and its effect on biomarkers of mineral and bone homeostasis, lower incidence of hypophosphatemia in the FDI group was observed [27]. Also Detlie et al. among IBD patients showed that ferric carboxymaltose was associated with a higher incidence of hypophosphatemia compared with iron derisomaltose [28]. Our results demonstrate and prove that both iron formulations are safe and effective in the treatment of anaemia in IBD patients. 

Thirdly, in our study we have observed a statistically significant increase of haemoglobin, MCHC, iron and ferritin levels (*p* < 0.001) after iron repletion. That observation was related to the intended effect of the therapy as previously had been described in other studies [26].

This trial has its limitations: first of all, even though groups of patients were sufficient to evaluate effect of iron formulas on bone mineral alterations in IBD patients, may not be sufficient to compare effect of each formula on bone mineral alteration (FDI was applied in majority of patients, whereas when patients were allergic to it or did not respond to it, FCM was administered). This way most of patients received FDI and the group who received FCM was smaller. Secondly, this study was a real world study and patients were not randomly assigned to two groups, what may constitute a limitation. 

## 5. Conclusions

This study indicates that transient and non-significant disorders of phosphate metabolism were found after parenteral iron infusions in the studied group, which do not constitute a contraindication to treatment with parenteral iron preparations. We consider iron iv administration safe and efficient in IBD patients. 

## Figures and Tables

**Figure 1 jpm-13-00458-f001:**
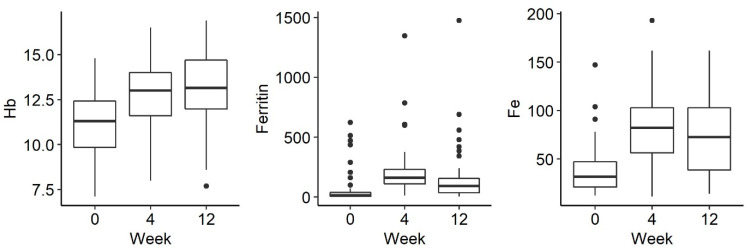
Changes in Hg (hemoglobin), ferritin and Fe (iron) levels over time: before iron administration and 4 and 12 weeks after administration.

**Figure 2 jpm-13-00458-f002:**
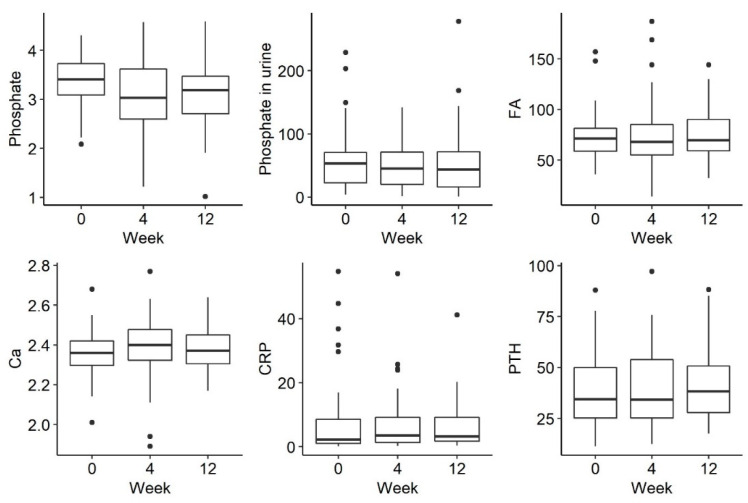
Changes in the concentration of phosphate, phosphate in urine, FA (alkaline phosphatase), Ca (calcium), CRP (C-reactive protein) and PTH (parathyroid hormone) before iron administration and 4 weeks and 12 weeks after administration.

**Figure 3 jpm-13-00458-f003:**
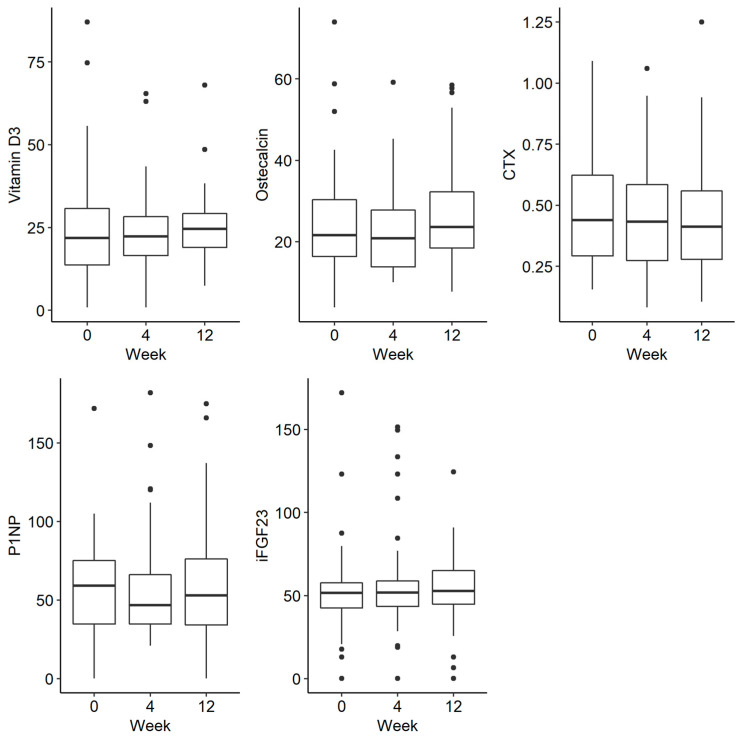
Changes in the concentration of vitamin D, osteocalcin, CTX (C-terminal telopeptide), P1NP (procollagen type 1N propeptide) and iFGF23 (intact fibroblast growth factor) before iron administration and 4 and 12 weeks after administration.

**Table 1 jpm-13-00458-t001:** Patient demographics at baseline.

	CD *n* = 32	UC *n* = 24
Female, *n* (%) Male, *n* (%)	12 (37.5%) 20 (62.5%)	12 (50%) 12 (50%)
Median age (years) (SD)	38 ± 14.99	35.35 ± 13.51
Disease activity	CDAI < 150: 14 patients CDAI 151–220: 12 patients CDAI 221–450: 6 patients CDAI > 450: 0 patient	Truelove Witts scale remission: 2 mild:12 moderate: 9 severe: 1
Current use of medications		
5-ASA (mesalasine or suphasalasine)	30 (93.8%)	23 (96%)
Corticosteroids Prednisone or methylprednisolone	4 (12.5%)	5 (20.8%)
Budesonide	3 (9.4%)	6 (25%)
Immunosuppressants		
Azathioprine	16 (50%)	9 (37.5%)
Mercaptopurine	2 (6.25%)	2 (8.3%)
Methotrexate	2 (6.25%)	
Biologic agents		
Infliximab	4 (12.5%)	
Adalimumab	4 (12.5%)	
Vedolizumab	1 (3.1%)	3 (1.25%)
Ustekinumab Etrolizumab		1(4.1%)

Abbreviations: n: number of patients, SD: standard deviation, CD: Crohn’s disease, UC: ulcerative colitis, 5-ASA: 5-aminosalicylates, CDAI: Crohn’s Disease Activity Index.

**Table 2 jpm-13-00458-t002:** The influence of iron formulations on the tested parameters in 3 measurements, including *post hoc* tests.

	N	DAY “0”			4 WEEKS AFTER IV Fe			12 WEEKS AFTER IV Fe			*p*
		Mean Rank	Me	IQR	Mean Rank	Me	IQR	Mean Rank	Me	IQR	
**Albumins**	34	1.63 _a_	4.29	0.49	2.18 _b_	4.36	0.47	2.19 _b_	4.40	0.50	0.031
**ALP**	36	1.79 _a_	71.50	23.50	2.24 _a_	68.00	32.50	1.97 _a_	69.50	32.50	0.155
**VITAMIN B_12_**	21	2.14 _a_	397.00	237.50	1.81 _a_	329.00	232.50	2.05 _a_	360.00	268.00	0.526
**CTX**	41	2.29 _a_	0.44	0.33	1.91 _a_	0.43	0.31	1.79 _a_	0.41	0.28	0.060
**CA**	42	1.77 _a_	2.36	0.11	2.00 _a_	2.38	0.15	2.23 _a_	2.38	0.14	0.106
**CRP**	30	1.72 _a_	4.00	9.53	2.18 _a_	3.65	7.83	2.10 _a_	2.80	893	0.151
**Ferritin**	37	1.09 _a_	1.00	22.50	2.78 _b_	155.00	121.00	2.12 _c_	91.00	14050	<0.001
**Folic acid**	20	2.25 _a_	7.20	9.05	2.03 _a_	9.55	9.28	1.73 _a_	6.70	9.63	0.245
**HB**	38	1.16 _a_	11.30	2.80	2.38 _b_	13.05	2.50	2.46 _b_	13.10	2.68	<0.001
**HCT**	38	1.22 _a_	36.90	7.80	2.43 _b_	41.70	5.55	2.34 _b_	40.65	8.43	<0.001
**MCV**	38	1.34 _a_	85.45	10.03	2.00 _b_	88.05	9.82	2.66 _c_	89.70	7.88	<0.001
**MCHC**	38	1.55 _a_	29.70	2.30	2.01 _b_	30.75	2.13	2.43 _b_	31.95	2.60	0.001
**Osteocalcin**	32	1.84 _a_	17.90	15.73	1.86 _a_	19.90	14.25	2.30 _a_	23.55	15.00	0.116
**PHOSPHATE IN SERUM**	41	2.30 _a_	3.38	0.65	1.88 _a_	3.00	0.99	1.82 _a_	3.19	0.76	0.054
**Phosphate in urine**	36	2.14 _a_	58.45	57.78	1.88 _a_	52.85	62.35	1.99 _a_	53.60	63.45	0.527
**PTH**	37	1.92 _a_	38.30	29.10	2.14 _a_	33.60	27.90	1.95 _a_	37.20	23.30	0.594
**FE**	38	1.17 _a_	29.00	22.00	2.58 _b_	81.50	50.50	2.25 _b_	73.00	65.25	<0.001
**TIBIC**	35	2.69 _a_	361.00	62.00	1.57 _b_	285.00	70.00	1.74 _b_	280.00	103.00	<0.001
**VIT D3**	33	1.94 _a_	19.20	18.35	1.86 _a_	21.90	12.60	2.20 _a_	24.60	14.50	0.359
**IFGF23**	41	1.88 _a_	51.53	17.05	1.98 _a_	49.64	18.63	2.15 _a_	54.49	22.13	0.461
**P1NP**	40	2.05 _a_	53.80	38.29	1.88 _a_	45.05	32.95	2.08 _a_	54.98	41.86	0.622

Note: Each letter a, b and c in the subscript represents a subset of categories whose proportions do not differ significantly from each other at *p* < 0.05. Abbreviations: ALP: alkaline phosphatase, CTX: C-terminal telopeptide, Ca: Calcium, CRP: C-reactive protein, Hb: hemoglobin, Hct: hematocrit, MCV: mean corpuscular volume, MCHC: mean corpuscular hemoglobin concentration, PTH: parathyroid hormone, Fe: iron, TIBC: total iron binding capacity, Vit D: vitamin D, iFGF23: intact fibroblast growth factor, P1NP: procollagen type 1N propeptide.

**Table 3 jpm-13-00458-t003:** Comparison of the biological indicators between patients using ferric derisomaltose and ferric carboxymaltose.

	Ferric Derisomaltose			Ferric Carboxymaltose			Z	*p*
	Mean Rank	Me	IQR	Mean Rank	Me	IQR		
Change between day 7 and baseline measurement								
Albumins	9.15	0.11	0.57	8.50	−0.14	0.57	−0.23	0.821
ALP	8.75	3.00	17.75	10.17	5.00	6.00	−0.44	0.659
Vitamin B12	3.00	−32.00	106.00	3.00	6.00	21.00	0.00	1.000
CTX	11.75	−0.05	0.21	16.21	0.04	0.18	−1.36	0.173
Ca	12.94	0.03	0.17	11.43	0.05	0.19	−0.48	0.633
CRP	10.67	0.40	5.80	11.83	3.76	47.75	−0.39	0.697
Ferritin	12.13	459.00	276.00	8.17	93.00	732.00	−1.32	0.186
Folate acid	1.50	−2.80	4.50	4.00	1.00	0.00	−1.73	0.083
Hb	12.35	0.70	1.65	12.86	1.00	1.50	−0.16	0.873
Hct	12.15	2.30	4.25	13.36	3.50	5.40	−0.38	0.703
MCV	12.85	1.60	3.15	11.64	2.20	5.20	−0.38	0.703
MCHC	12.12	−0.10	1.00	13.43	0.20	0.80	−0.41	0.679
Osteocalcin	12.33	−4.85	10.00	13.00	−5.75	14.85	−0.20	0.841
Phosphate	14.18	−0.32	1.21	8.43	−0.99	0.75	−1.81	0.070
Phosphate in urine	12.27	−4.10	33.80	9.86	−12.70	177.30	−0.81	0.418
PTH	14.56	−1.85	23.25	9.00	−22.40	44.50	−1.69	0.090
Fe	9.13	53.00	54.00	13.25	117.00	149.50	−1.30	0.193
TIBIC	9.67	−32.00	101.00	11.25	−3.50	103.00	−0.50	0.617
Vitamin D3	6.00	−0.75	2.40	8.60	−0.20	1.80	−1.17	0.241
iFGF23	11.59	12.01	40.60	13.17	21.74	114.16	−0.49	0.624
P1NP	10.40	−9.90	26.37	12.50	−5.20	23.69	−0.70	0.484
**Measurement 7th day**								
Albumins	9.23	4.35	0.37	8.25	4.19	1.02	−0.34	0.734
ALP	8.50	68.00	36.00	11.33	70.00	−66.00	−0.88	0.377
Vitamin B12	4.50	548.50	-475.00	2.00	338.00	−291.00	−1.73	0.083
CTX	12.17	0.27	0.44	15.14	0.35	0.25	−0.91	0.364
Ca	13.74	2.41	0.18	9.50	2.23	0.27	−1.34	0.182
CRP	10.40	3.30	9.20	12.50	6.75	5.48	−0.70	0.483
Ferritin	12.13	482.00	238.00	8.17	145.50	840.50	−1.32	0.186
Folate acid	1.50	2.90	−2.80	4.00	3.90	−380	−1.73	0.083
Hb	12.38	11.70	2.95	12.79	11.60	2.90	−0.13	0.899
Hct	12.38	38.80	8.05	12.79	36.70	8.80	−0.13	0.899
MCV	13.06	89.50	10.65	11.14	85.60	7.90	−0.60	0.546
MCHC	12.76	30.40	1.35	11.86	30.70	2.40	−0.29	0.775
Osteocalcin	11.53	17.60	10.73	15.42	22.25	19.58	−1.17	0.243
Phosphate	13.65	2.95	1.25	9.71	2.65	1.63	−1.24	0.216
Phosphate in urine	11.19	23.10	33.80	13.86	54.20	77.40	−0.87	0.385
PTH	14.39	36.50	29.15	9.43	29.10	26.00	−1.51	0.130
Fe	9.53	110.00	63.00	11.75	162.50	183.50	−0.70	0.484
TIBIC	10.03	306.00	50.00	9.88	308.00	100.50	−0.05	0.960
Vitamin D3	7.25	19.40	24.80	6.60	14.20	20.55	−0.29	0.770
iFGF23	12.44	64.40	38.52	12.67	61.98	118.35	−0.07	0.947
P1NP	11.67	44.65	27.50	15.00	55.75	59.98	−1.00	0.317

Abbreviations: Me: median value, IQR: interquartile range, Z: Z-score, ALP: alkaline phosphatase, CTX: C-terminal telopeptide, Ca: Calcium, CRP: C-reactive protein, Hb: hemoglobin, Hct: hematocrit, MCV: mean corpuscular volume, MCHC: mean corpuscular hemoglobin concentration, PTH: parathyroid hormone, Fe: iron, TIBC: total iron binding capacity, iFGF23: intact fibroblast growth factor, P1NP: procollagen type 1N propeptide.

**Table 4 jpm-13-00458-t004:** Comparison of changes in biological markers in patients receiving ferric derisomaltose and ferric carboxymaltose.

	Change *	Ferric Derisomaltose (*n* = 39)			Ferric Carboxymaltose (*n* = 11)			Z	*p*
		Mean Rank	Me	IQR	Mean Rank	Me	IQR		
Albumins	1	21.18	0.17	0.34	25.11	0.17	0.62	−0.84	0.03
	2	21.04	0.07	0.40	27.45	0.31	0.56	−1.39	0.166
ALP	1	23.28	1.00	14.50	21.69	0.50	23.00	−0.31	0.755
	2	23.71	3.50	22.50	22.75	1.50	25.00	−0.20	0.842
Vitamin B12	1	17.54	−17.00	162.00	17.36	−60.00	167.00	−0.04	0.966
	2	19.98	5.00	111.00	14.06	−95.00	166.50	−1.46	0.144
CTX	1	23.79	−0.02	0.25	29.70	0.04	0.32	−1.17	0.244
	2	21.47	−0.04	0.18	24.00	−0.04	0.45	−0.54	0.591
Ca	1	24.21	0.01	0.13	30.09	0.10	0.20	−1.18	0.236
	2	22.73	0.02	0.15	28.70	0.09	0.12	−1.22	0.221
CRP	1	23.44	0.55	3.23	17.82	0.20	34.30	−1.28	0.200
	2	19.77	0.40	7.40	18.29	0.70	37.30	−0.32	0.749
Ferritin	1	22.51	132.00	132.00	29.50	196.50	227.00	−1.43	0.153
	2	21.85	54.50	116.50	24.70	87.00	251.50	−0.62	0.538
Folate acid	1	16.75	−0.10	5.08	18.40	0.00	3.10	−0.35	0.725
	2	16.58	−0.55	10.85	14.00	−1.00	3.30	−0.66	0.508
Hb	1	25.49	1.85	1.75	23.32	1.60	2.30	−0.44	0.657
	2	22.59	1.90	2.40	22.17	1.90	1.35	−0.09	0.930
Hct	1	24.64	3.90	4.33	26.23	4.80	4.90	−0.32	0.746
	2	22.69	4.50	6.00	21.78	3.90	2.70	−0.19	0.850
MCV	1	24.80	3.25	4.63	25.68	3.60	6.10	−0.18	0.857
	2	22.59	5.40	7.70	22.17	4.40	5.25	−0.09	0.930
MCHC	1	26.22	1.20	2.70	20.77	0.10	2.20	−1.11	0.265
	2	22.11	1.10	2.80	24.00	1.30	4.65	−0.39	0.694
Osteocalcin	1	21.22	−1.40	9.80	22.40	0.15	18.0	−0.27	0.790
	2	21.71	3.00	13.68	20.63	−0.10	25.90	−0.22	0.823
Phosphate	1	26.51	−0.02	0.68	19.10	−0.31	1.70	−1.46	0.143
	2	24.42	−0.25	0.64	22.45	−0.25	1.46	−0.40	0.687
Phosphate in urine	1	22.71	−6.10	38.48	21.80	−6.85	47.08	−0.20	0.845
	2	21.72	−4.10	67.15	17.27	−17.60	71.40	−1.08	0.282
PTH	1	24.07	0.40	14.35	25.95	11.40	32.50	−0.39	0.695
	2	22.67	0.40	17.95	17.22	−5.30	35.15	−1.18	0.238
Fe	1	22.22	33.50	47.00	29.82	58.00	61.00	−1.61	0.108
	2	22.78	35.50	55.50	26.10	46.00	71.00	−0.69	0.488
TIBIC	1	21.59	−82.00	107.00	27.95	−43.00	86.00	−1.35	0.176
	2	21.89	−77.00	90.00	26.90	−39.50	56.50	−1.06	0.287
Vitamin D3	1	21.34	−0.15	7.95	19.78	−1.40	5.60	−0.35	0.729
	2	21.05	2.90	16.70	18.85	4.20	7.20	−0.52	0.606
iFGF23	1	25.10	0.97	11.04	24.60	−1.84	58.10	−0.10	0.921
	2	24.71	2.18	11.72	19.64	−0.94	26.93	−1.09	0.274
P1NP	1	23.43	−3.00	35.69	23.78	−3.78	51.46	−0.07	0.945
	2	23.23	0.34	23.79	22.20	−0.70	73.53	−0.22	0.827

Abbreviations: *—change 1 (between measurement in the week 4 and week 0) and 2 (change between week 12 and week 0). ALP: alkaline phosphatase, CTX: C-terminal telopeptide, Ca: Calcium, CRP: C-reactive protein, Hb: hemoglobin, Hct: hematocrit, MCV: mean corpuscular volume, MCHC: mean corpuscular hemoglobin concentration, PTH: parathyroid hormone, Fe: iron, TIBC: total iron binding capacity, iFGF23: intact fibroblast growth factor, P1NP: procollagen type 1N propeptide.

**Table 5 jpm-13-00458-t005:** Relationship of phosphate and iFGF23 levels in individual measurements.

			MEASUREMENT “0”		
			Phosphate	Phosphate in Urine	iFGF23
MEASUREMENT “0”	Phosphate in urine	*r*	−0.09	–	–
		*p*	0.531		
	iFGF23	*r*	0.23	−0.13	–
		*p*	0.095	0.359	
MEASUREMENT AFTER 4 WEEKS	Phosphate	*r*	0.50	−0.17	0.14
		*p*	<0.001	0.249	0.351
	Phosphate in urine	*r*	0.01	0.29	−0.05
		*p*	0.933	0.056	0.768
	iFGF23	*r*	0.25	−0.08	0.50
		*p*	0.081	0.592	<0.001
MEASUREMENT AFTER 12 WEEKS	Phosphate	*r*	0.46	0.08	−0.02
		*p*	0.001	0.606	0.893
	Phosphate in urine	*r*	−0.16	0.31	−0.08
		*p*	0.307	0.054	0.628
	iFGF23	*r*	0.17	0.02	0.57
		*p*	0.245	0.892	<0.001

A moderate correlation of phosphate concentration and a strong correlation of iFGF23 were observed in subsequent measurements – in weeks 4 and 12—using Pearson’s *r* correlation analysis.

**Table 6 jpm-13-00458-t006:** The relationship of phosphorus and iFGF23 levels in individual measurements, broken down by drug groups.

				MEASUREMENT “0”		
				Phosphate	Phosphate in Urine	IFGF23
Ferric derisomaltose	MEASUREMENT “0”	Phosphate in urine	*R*	−0.11	-	-
			*p*	0.499		
		iFGF23	*R*	0.33	−0.09	-
			*p*	0.033	0.576	
	MEASUREMENT AFTER 4 WEEKS	Phosphate	*R*	0.50	−0.25	0.19
			*p*	0.001	0.134	0.259
		Phosphate in urine	*r*	−0.08	0.17	0.01
			*p*	0.658	0.337	0.964
		iFGF23	*r*	0.28	−0.01	0.63
			*p*	0.082	0.946	<0.001
	MEASUREMENT AFTER 12 WEEKS	Phosphate	*r*	0.37	−0.07	0.13
			*p*	0.026	0.718	0.438
		Phosphate in urine	*r*	−0.22	0.36	−0.03
			*p*	0.233	0.055	0.879
		IFGF23	*r*	0.29	0.02	0.68
			*p*	0.085	0.899	<0.001
Ferric carboxymaltose	MEASUREMENT “0”	Phosphate in urine	*r*	−0.14	-	-
			*p*	0.661		
		iFGF23	*r*	−0.35	−0.30	-
			*p*	0.289	0.365	
	MEASUREMENT AFTER 4 WEEKS	Phosphate	*r*	0.72	−0.01	−0.13
			*p*	0.019	0.978	0.733
		Phosphate in urine	*r*	0.30	0.60	−0.37
			*p*	0.407	0.065	0.333
		iFGF23	*r*	0.17	−0.22	0.01
			*p*	0.648	0.536	0.989
	MEASUREMENT AFTER 12 WEEKS	Phosphate	*r*	0.67	0.26	−0.41
			*p*	0.035	0.474	0.240
		Phosphate in urine	*r*	0.10	0.32	−0.45
			*p*	0.778	0.341	0.169
		iFGF23	*R*	−0.01	0.13	0.16
			*p*	0.975	0.707	0.641

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
