# Peer review of "Effect of Intravenous Iron Administration on Bone Mineral and Iron Homeostasis in Patients with Inflammatory Bowel Disease—Results of a Prospective Single-Centre Study"

_jpm, 2023, doi:10.3390/jpm13030458_

Round 1

Reviewer 1 Report

-        “parenteral symptomatology”

What’s the meaning of “parenteral symptomatology”?

-        “its prevalence affects 45% of the IBD population”

Don’t give a so specific number

-        “Administered intravenously, iron is more effective, delivers a faster response and is better tolerated than oral iron.”

This is a debated topic, see: “Vernero M et al. Oral iron supplementation with Feralgine® in inflammatory bowel disease: a retrospective observational study. Minerva Gastroenterol Dietol. 2019 Sep;65(3):200-203. doi: 10.23736/S1121-421X.19.02572-8. Epub 2019 Mar 27. PMID: 30917629.”

-        “Of the 56 patients”

In table 1 there are 33 CD + 24 UC = 57 patients

-        Report in Table 2, 3, 4 median or mean values according to normal distribution and not average ranks; explain the abbreviation of the Table (for example, Me in Table 3)

-        What about the dosage of steroids during the study and its influence on the measured values?

-        Report the adverse events with the two different formulation

Author Response

Dear Sir/Madam,  

Thank you for all valuable comments on this article. According to your revision let me comment point by point general thoughts about my work:

 According to “parenteral symptomatology” I meant extraintestinal symptoms, and I have already changed it to a more appropriate expression. 

According to data from other works the prevalence may affect almost half  of the IBD population, but the data is variable. I have cited few of them.

 “Administered intravenously, iron is more effective, delivers a faster response and is better tolerated than oral iron.” This is a debated topic, see: “Vernero M et al. Oral iron supplementation with Feralgine® in inflammatory bowel disease: a retrospective observational study. Minerva Gastroenterol Dietol. 2019 Sep;65(3):200-203. doi: 10.23736/S1121-421X.19.02572-8. Epub 2019 Mar 27. PMID: 30917629.”

 In this cited retrospective study by Vernero the studied population was quite small ( n=52, of course in some cases oral treatment may be useful, however in most IBD patients knowing the fact that inflammation may be localized in the small intestine and according to possible reaction of Fenton, IV iron may be better tolerated. I have changed it as followed:

“Even though there is some data showing that in mild or moderate anaemia in IBD oral formulations may be useful, in general administered intravenously iron is more effective, delivers a faster response and is better tolerated than oral iron”.

In table 1 there are 33 CD + 24 UC = 57 patients. I am sorry for that error- in total there were 56 patients what is shown in the Table nr 1.

Report in Table 2, 3, 4 median or mean values according to normal distribution and not average ranks; explain the abbreviation of the Table (for example, Me in Table 3)

Thank you.

let me explain that: both tests - U Mann Whitney and H Kruskal Wallis - are non-parametric tests, which means that the significance of the difference between the groups is based on ranks and not on means as in Student's t-test or in ANOVA.  I have also explained abbreviations in the table 3 and 4 which were missing.

What about the dosage of steroids during the study and its influence on the measured values?

 Even though in the patients demographics different types of steroids were detailed, its relationship with iv iron wasn’t analyzed afterwards (it was not the topic of this study). We may think about it in the next study

Report the adverse events with the two different formulation

In our study we have excluded patients who during the iron infusion presented serious adverse events, not  receiving full dose if iron IV.

Thank you once again for your valuable comments

Yours faithfully,

E.Tulewicz-Marti

Reviewer 2 Report

The article entitled “Intravenous iron administration in patients with inflammatory bowel disease– friend or foe in bone mineral alterations: Results of a prospective single-centre study” has been prepared in an acceptable way, but some minor concerns were risen. 

Strengths
This research article by Tulewicz-Marti et al. focused on the effect of parenteral iron administration on biomarkers of mineral and bone homeostasis over time. This topic is important because it addresses the treatment of inflammatory bowel disease patients via parenteral iron. This is very important as this is having lesser side effects and a new therapeutic strategy for this disease condition.

Limitations:
Title is not convincing. I would suggest to rewrite it. This paper focused on the 
investigation of changes in calcium and phosphate levels in blood and urine, iFGF 23 after iron infusions of carboxymaltose and derisomaltose in adults with IBD and they ignored the other important parameters.

Minor concerns:

1.       It is not clear in the table what is the % data in row 1 they mentioned?

2.       In Table 2, why did the N show low sample size only in case of vitamin B12 and folic acid 

3.       In Table 3, use upper case for average rank at both places.

4.       Could please explain the average rank in Table 3? 

5.       Try to make colored graphs for these all parameters that will look great.

6.       Why did author target to measure the level of iFGF 23 after traetment?

7.       Why did author consider only blood-related parameters? Are there any human small intestine/colon biopsy samples collected for particularly see the effect of this treatment in IBD patients?

8.       Please try to clarify the statics.

9.       Check the sentence chi square and Pearson’s r correlation. P < 0.05 was considered statistically significant in statistical analysis portion for font size. 

10.    Please check the references style and add some recent references related to this study.

11.    Please improve the grammar. Read carefully the full manuscript for typos and grammar.

Author Response

Dear Sir/Madam,

Thank you for the review and the comments. Below I am presenting explanation to revisions and changes made to the manuscript.

According to the 1st revision:

It is not clear in the table what is the % data in row 1 they mentioned?

Thank you for that comment. I have modified this data showing the number of female and male patients and also the percentage of each group

In Table 2, why did the N show low sample size only in case of vitamin B12 and folic acid 

According to the low sample size of vitamin B12 and folic acid not all patients had this two parameter’s determined.

In Table 3, use upper case for average rank at both places.

Thank you- I have corrected it in the manuscript.

 Could please explain the average rank in Table 3? 

By average rank I meant mean rank – this expression is better in this context.

Why did author target to measure the level of iFGF 23 after treatment?

According to data from the literature iFGF23 may be elevated after parenteral iron infusions, especially after some formulas such as FCM. Therefore we have decided to measure iFGF 23 and evaluate its correlation with hipopophosphatemia.

Why did author consider only blood-related parameters? Are there any human small intestine/colon biopsy samples collected for particularly see the effect of this treatment in IBD patients?

According to the protocol of the study we have considered mostly serum parameters (also phosphate in urine). We weren’t going to collect the colon biopsies in this study.

Please try to clarify the statics.

BM SPSS Statistics 25 was used to answer the research question. Descriptive statistics, together with Kolmogorov-Smirnov tests, were calculated.   The U Mann–Whitney test, Friedman test, chi-square test of independence and Pearson' r correlation was performed. P < 0.05 was considered statistically significant.

Check the sentence chi square and Pearson’s r correlation. P < 0.05 was considered statistically significant in statistical analysis portion for font size. 

I have corrected it, thank you

Please check the references style and add some recent references related to this study.

I have already corrected this  and updated the references

Please improve the grammar. Read carefully the full manuscript for typos and grammar.

Yours faithfully,

E.Tulewicz-Marti

Round 2

Reviewer 1 Report

Thak you for the corrections.